# Toxicity of TiO_2_, ZnO, and SiO_2_ Nanoparticles in Human Lung Cells: Safe-by-Design Development of Construction Materials

**DOI:** 10.3390/nano9070968

**Published:** 2019-07-02

**Authors:** Monika Remzova, Radek Zouzelka, Tana Brzicova, Kristyna Vrbova, Dominik Pinkas, Pavel Rőssner, Jan Topinka, Jiri Rathousky

**Affiliations:** 1J. Heyrovsky Institute of Physical Chemistry of the CAS, Dolejskova 3, 18223 Prague, Czech Republic; 2Institute of Experimental Medicine of the CAS, Videnska 1083, 14220 Prague, Czech Republic; 3Institute of Molecular Genetics of the CAS, Microscopy Center, Electron Microscopy Core Facility, 14220 Prague, Czech Republic

**Keywords:** SiO_2_ nanoparticles, ZnO nanoparticles, TiO_2_ nanoparticles, toxicity, ethyl silicate consolidants

## Abstract

Rapid progress in the development of highly efficient nanoparticle-based construction technologies has not always been accompanied by a corresponding understanding of their effects on human health and ecosystems. In this study, we compare the toxicological effects of pristine TiO_2_, ZnO, SiO_2_, and coated SiO_2_ nanoparticles, and evaluate their suitability as additives to consolidants of weathered construction materials. First, water soluble tetrazolium 1 (WST-1) and lactate dehydrogenase (LDH) assays were used to determine the viability of human alveolar A549 cells at various nanoparticle concentrations (0–250 μg mL^−1^). While the pristine TiO_2_ and coated SiO_2_ nanoparticles did not exhibit any cytotoxic effects up to the highest tested concentration, the pristine SiO_2_ and ZnO nanoparticles significantly reduced cell viability. Second, as all developed nanoparticle-modified consolidants increased the mechanical strength of weathered sandstone, the decisive criterion for the selection of the most suitable nanoparticle additive was as low toxicity as possible. We believe that this approach would be of high importance in the industry, to identify materials representing top functional properties and low toxicity, at an early stage of the product development.

## 1. Introduction

Nanoparticles are widely used because they can improve both the quantitative and qualitative properties of technological materials [1]. In the construction industry, they can enhance the mechanical properties of the materials used to consolidate weathered building materials [2]. However, their production, handling and use can pose health and environmental risks that potentially limit their benefits.

From the application point of view, SiO_2_, ZnO, and TiO_2_ form a triad of the most commonly used oxide nanoparticles. While their surface properties are comparable, their structural properties differ substantially, particularly their solubility and conductivity. While TiO_2_ is an insoluble ceramic and SiO_2_ is practically insoluble, ZnO displays considerable solubility in weak acids and, thus, suffers from photocorrosion [3]. ZnO and TiO_2_ are semiconductors with a similar band-gap [4,5,6,7], whereas SiO_2_ is an insulant. 

These properties are however, associated with a broad range of toxicological effects. In terms of solubility, practically insoluble SiO_2_ nanoparticles have been shown to induce reactive oxygen species (ROS) autophagy in human hepatocytes [8], spermatogenesis damage [9], and impairment of vascular homeostasis [10]. Soluble ZnO nanoparticles release toxic zinc ions, prior to or after their uptake into cells; their toxic effects exhibit a sharp concentration dependence, indicating the presence of a critical Zn^2+^ ion concentration [11]. Overtreatment with zinc ions causes a cell to lose its control functions and commit apoptosis [12]. In terms of semiconductivity, the toxicity of zinc oxide nanoparticles can be increased by the photocatalytic effect [13] In the case of insoluble TiO_2_ semiconductor nanoparticles, they do not appear to become toxic under dark or non-ultraviolet (UV) exposure conditions [14], but toxicity has been reported after UV exposure [15].

From the above, it is obvious that when developing any nanoparticle-based technology, the toxicological impact of nanoparticles needs to be taken into account [16]. However, this aspect is generally neglected, with nanoparticles often selected only on the basis of their functions. Consequently, in many cases, the developed technologies are not safe-by-design.

Therefore, in this study, we developed safe-by-design, highly efficient consolidants for weathered construction materials. First, we assessed the cytotoxicity of five types of oxide nanoparticles (TiO_2_, ZnO, SiO_2_, and octyl- and methyl-modified SiO_2_). Second, based on the results of two different cytotoxicity tests, we formulated nanoparticle-modified consolidants with an optimized composition to achieve high performance characteristics with minimum health hazards. Our research shows that the frequently used toxic nanoparticles can be replaced by non-toxic equivalents without any performance impairment.

## 2. Materials and Methods 

### 2.1. Nanoparticles and Their Characterization

The following five commercially available metal oxide nanoparticles were tested: 

Aeroxide^®^ TiO_2_ P25 (Evonik Industries, Essen, Germany); Aerosil^®^ SiO_2_ 200 Pharma, R805, R9200 (Evonik Industries, Essen, Germany) and NanoZnO (Bochemie, Bohumín, Czech Republic).

Aerosil^®^ SiO_2_ A200 (hereinafter referred to as SiO_2_), which is a nanopowder with a surface area of 220 m^2^ g^−1^ and particle size of 12 nm (both specifications are that provided by the manufacturer).Aerosil^®^ R9200 (hereinafter referred to as SiO_2_–methyl), which is a nanopowder of SiO_2_ with a methylated surface, its surface area and particle size were 150–190 m^2^ g^−1^ and 12 nm (as stated by the manufacturer), respectively.Aerosil^®^ R805 (hereinafter referred to as SiO_2_–octyl), which is a nanopowder of SiO_2_ with an octylated surface, its surface area and particle size was 125–175 m^2^ g^−1^ and 12 nm (as stated by the manufacturer), respectively.Aeroxide^®^ TiO_2_ P25 (hereinafter referred to as TiO_2_), which is a photocatalyst that is widely used, owing to its high activity in many photocatalytic reactions. It contains more than 70% of the anatase phase with a minor proportion of rutile (about 20%) and a small percentage of the amorphous phase [17]. It exhibits a specific surface area in the range of 35–65 m^2^ g^−1^ and a particle size of 25 nm (as stated by the manufacturer).NanoZnO (hereinafter referred to as ZnO), which is a photocatalyst with a surface area and particle size of 90–110 m^2^ g^−1^ and 17 nm (as stated by the manufacturer), respectively.

The morphological properties of the nanoparticles were determined by the analysis of adsorption isotherms of nitrogen or krypton at ca 77 K, using a Micrometrics 3Flex volumetric adsorption unit, by scanning electron microscopy (SEM),using JSM-6700F microscope (Jeol, Tokyo, Japan) and by high-resolution transmission electron microscopy (HRTEM) using JEM-2100Plus instrument (Jeol, Tokyo, Japan). The size distribution and zeta potential of the nanoparticles in dispersions (in water-bovine serum albumin (BSA) and cell culture medium) was measured by dynamic light scattering using ZetaSizer Nano ZS (Malvern Instruments Ltd., Malvern, UK). The structural properties of the nanoparticles were determined by Fourier transform infrared (FTIR) spectroscopy using a Nicolet 6700 apparatus (Madison, WI, USA).

### 2.2. Preparation of the Nanoparticle Dispersions for Cytotoxicity Testing

To disperse hydrophobic nanoparticles in water-based systems, they were first treated with ethanol (70 %). The nanoparticles were then dispersed in deionized water (2.56 mg mL^−1^) containing 0.05% BSA and sonicated by probe at 400 W, with an amplitude of 10% (Digital Sonifier S-450d equipped with a standard 13-mm disruptor horn, Branson Ultrasonics, Danbury, CT, USA) for 16 minutes, in an ice bath. Prior to exposure to the cells, the nanoparticle dispersions were gradually diluted in the cell culture medium containing 1% fetal bovine serum (FBS), to their final concentrations of 1, 5, 10, 25, 50, 100, 150, and 250 µg mL^−1^. 

### 2.3. Cell Cultivation and Exposure to the Nanoparticles

The A549 cell line (human type II pulmonary epithelial cells, CCL-185™ ATCC) was cultured in a minimal essential medium (MEM) + Glutamax (Gibco™, Thermo Fisher Scientific Grand Island, NY, USA) and FBS of 10% (v/v) (Gibco™, Sigma-Aldrich, St. Louis, MO, USA) in an incubator (37 °C, 5% CO_2_). For the cytotoxicity testing, the cells were seeded overnight in 96-well microtiter plates with 7,500 cells per well and were incubated overnight. Freshly prepared nanoparticles at the above given concentrations were added to the wells and incubated for 24 h (37 °C, 5% CO_2_)

### 2.4. Cytotoxicity Testing

Water soluble tetrazolium 1 (WST-1) assay: After the exposure period, the cell culture medium was removed and the cells were twice-rinsed with Phosphate Buffered Saline (PBS). The Cell Proliferation Reagent WST-1 (Roche Diagnostics, Mannheim, Germany) and a phenol-red-free MEM containing 1% PBS was mixed in a ratio of 1:10. A total of 120 μl of this mixture was then added to each test well and incubated at 37 °C for 1 h. To prevent the interference of nanoparticles being adsorbed on the plastic plate with the absorbance reading, 100 μL of supernatants from each well were transferred to a new plate. The absorbance of the well content was measured at the wavelength of 450 nm using a SpectraMax^®^ M5 Plate Reader (Molecular Devices, Sunnyvale, CA, USA). To determine the cell viability, the background absorbance of the well content without cells, was subtracted. The viability of the nanoparticle-treated lung cells was expressed as a ratio of the sample absorbance (Abs sample) and that of average negative control (100% viable, designated as average Abs NC)
% viability = (Abs sample/average Abs NC ) × 100(1)

As Zn^2+^ ions can be released from the ZnO nanoparticles, we employed the WST-1 assay with ZnCl2 to determine their cytotoxicity effect. The exposure concentrations were adjusted to provide the same dose of elemental Zn.

Lactate Dehydrogenase (LDH) assay: After 24 h incubation, 50 µL supernatant from each well was used to determine the released LDH activity (LDH_supernatants_).

The viable cells were washed with PBS and incubated with 100 µL of a Triton X-100 solution in a cell culture medium (1% wt., Sigma-Aldrich) at 37 °C for 30 min. A total of 50 µL of the supernatant from each well was used for the measurement of the LDH activity of the viable cells (LDH_lysates_). 

A total of 50 μL of the reaction mixture of the Cytotoxicity Detection Kit (LDH) (Roche Diagnostics, Mannheim, Germany) was added to both the above prepared supernatants and incubated in the dark for 15 min. Finally, 25 µL of 10 nM HCl was added to each well to terminate the reaction, and the absorbance at 490 nm was measured using the SpectraMax^®^ M5 Plate Reader. To determine the viability, the background absorbance of the well content without the cells was subtracted. The cell viability was calculated according to the following formula.
% viability = LDH_lysates_/(LDH_lysates_ + LDH_supernatants_) × 100(2)

Possible interference of the nanoparticles with the LDH assay was investigated by the incubation of two highest nanoparticle concentrations (100 and 250 µg mL^−1^) with cell lysates for 1 and 24 h before performing the LDH assay. No significant changes were detected in the absorbance values (representing the LDH activity).

### 2.5. Data Analysis and Statistics 

The toxicity experiments were done in three replicates for all treatments. Significant differences between the compared samples were determined using Student’s t test where p values were used as the threshold for statistical significance. An analysis of variance (ANOVA) followed by a Dunnett’s test was performed (The Prism 5 program, GraphPad Software, San Diego, CA, USA). Data are expressed as mean ± a standard deviation (SD). LC_50_ values (concentrations that inhibited cell viability by 50%) were calculated using four-parameter log-logistic models in the drc package in the statistical software R (version 3.4.0.) [18].

### 2.6. Nanoparticle Uptake by the Cells Determined by Transmission Electron Microscopy 

The cultivation of the cells in the presence of the nanoparticle dispersion (10 µg ml^−1^) on glass coverslips of 12 mm (Schott Glass AG) was carried out in the wells of a 24-well plate. After an initial 24 h exposure, the cells were washed with Sörensen buffer (0.1 M sodium/potassium phosphate buffer, pH 7.3; designated as SB), then treated with a 2.5% glutaraldehyde solution in SB for 2 h. These were then washed with SB and finally were treated with a 1% OsO_4_ solution in SB, for 2 h. Subsequently, the cells were dehydrated in acetone and embedded in Epon-Durcupan resin, which polymerized within 72 h at 60 °C. Ultrathin sections (80 nm) were placed on 200 mesh size copper grids and stained with uranyl acetate and lead citrate. The visualization was performed using a FEI Morgagni 268 transmission electron microscope operated at 80 kV, equipped with a Mega View III CCD camera (Olympus Soft Imaging Solution GmbH, Münster, Germany).

### 2.7. Preparation of the Nanoparticle-Modified Consolidants

The consolidants were prepared by adding the nanoparticles (3 wt. %) (see above) to Dynasylan^®^40 (Evonik, Essen, Germany), which is an ethylsilicate oligomer. Catalyst n-octylamine (0.18 wt. %) (Alfa Aesar, Ward Hill, MA, USA) was then added. The mixture obtained was diluted with isopropanol, in the ratio of 1:1. For the consolidant containing ZnO nanoparticles, a mixture of n-octylamine and dibutyltin dilaurate (DBTL) was used, as n-octylamine was not sufficient to achieve a formation of the gel. An overview of the consolidants is given in Table 1. Here, SiGel designated the consolidant containing Dynasylan®40, catalyst and the given nanoparticles. The commercial consolidant KSE OH (Remmers, Löningen, Germany) was used as a reference. Compared to our developed consolidants, KSE OH contained only 25% of solvents, while ours contained 50%. According to the data sheet, this product should be suitable for the consolidation of weathered, friable natural stones, particularly sandstones, cast stone, renders, and mortar. 

### 2.8. Evaluation of the Consolidation Effect

Naturally weathered sandstone (Prosek Rocks, Prague, Czech Republic) of high porosity and low strength was selected as a material that models a highly weathered construction material. This sandstone is an ocher colored fine-grained clastic sedimentary rock. Its mineral composition consists of 93% quartz clasts and 7% clay matrix (kaolinite, often impregnated with iron-oxyhydroxides). Blocks of the sandstone, 3 × 3 × 3 cm in size, were impregnated with the consolidants by capillary soaking, until the complete filling of the sandstone porous system. After six months, the mechanical properties of the treated stones were determined using a drilling resistance measuring system (DRMS) (SINT Technology, Calenzano, Italy). Drill bits of 4.8 mm diameter at a rotation speed of 300 rpm and a penetration rate of 30 mm/min were used.

## 3. Results and Discussion

### 3.1. Physico-Chemical Properties of the Nanoparticles

For the nanoparticles in powder form, nanoparticle aggregation was observed (Figure 1). The SiO_2_ nanoparticles formed large, fluffy, highly porous aggregates, while the coated nanoparticles were smaller, especially the methylated ones. The aggregates of the TiO_2_ nanoparticles were much smaller and rather inhomogeneous. Compared to the spherical character of the above-mentioned nanoparticles, the ZnO nanoparticles exhibited a rather different shape of platelets, which is clearly due to their crystallinity. The insets in Figure 1 show details of the particle structure and morphology. While amorphous SiO_2_ nanoparticles, both uncoated and coated, exhibit a characteristic spherical shape, highly crystalline TiO_2_ and ZnO nanoparticles differ in their morphology, being prismatic and platelet-like, respectively. The size of the uncoated and coated SiO_2_ and TiO_2_ nanoparticles was comparable—about 15–20 nm. However, compared to SiO_2_ nanoparticles, the ZnO particles were about 10 nm smaller and more aggregated. 

The size of the primary particles calculated from the Brunauer–Emmett–Teller (BET)surface area determined from the nitrogen sorption isotherms (Figure 2a), provided the crystals were approximated by a sphere, was in agreement with those determined from HRTEM images (Table 2, Figure 1). For SiO_2_ nanoparticles, the calculated primary size was in the range as that determined from HRTEM because the aggregates were very loose with complete accessibility of the surface. However, for TiO_2_ and ZnO the calculated values overestimated the particle size obtained by HRTEM, as these particles formed aggregates that were more compact.

Moreover, the *C* constant calculated from the BET equation, showed substantial differences in the surface properties of the nanoparticles. This constant was proportional to e^(q1−qL)/RT^, where *q_1_* is the adsorption heat in the first layer, *q_L_* is the liquefaction heat of the adsorption, *R* is the universal gas constant, and *T* is the absolute temperature. Consequently, it expresses the strength of the interaction between the adsorptive gas molecules and the surface. This strength was rather low for the organically-coated surfaces, especially for the octylated one (~27). This was due to an almost complete covering of the SiO_2_ surface with organic groups. However, regarding the uncoated inorganic surfaces, the *C* constant increased in the sequence SiO_2_–TiO_2_–ZnO, which was in agreement with the increasing density of these oxides.

In the cell culture medium, the nanoparticles exhibited very good stability with the exception of the SiO_2_ nanoparticles that formed clusters. Due to surface modification, the coated SiO_2_ were more stable and no larger clusters were formed. The zeta potential of all tested nanoparticles ranged from −14 to −20 mV. This indicated that the bovine serum albumin, the most abundant protein in serum (used as a dispersant) was adsorbed on the nanoparticle surface. 

The difference of crystallinity between the nanoparticles was significant. While the semiconductor TiO_2_ and ZnO nanoparticles exhibited a high degree of crystallinity, the coated and uncoated SiO_2_ did not (Figure 2b). The pattern of TiO_2_ P25 showed diffractions at 25.15° (101), 36.82° (103), 37.67° (004), 38.48° (112), 47.93° (200), 53.75° (105), and 54.96° (211), corresponding to the tetragonal anatase structure (space group I41/amd). Those at 27.33° (110), 35.93° (101), 41.18° (111), 54.21° (211), and 56.53° (220) were assigned to the tetragonal rutile structure (space group P42/mnm). From the diffractograms, it followed that the proportion of anatase and rutile phases was approximately 4:1. The diffractions of ZnO matched those of the hexagonal wurtzite structure (space group P63mc). Its pattern was characterized by the diffractions centered at 31.62° (100), 34.22° (002), 36.11° (101), 47.39° (102), and 56.47° (110). On the other hand, compared to highly crystalline TiO_2_ and ZnO, the coated and uncoated SiO_2_ nanoparticles showed an amorphous character. The characteristic asymmetrical peak centered at ca. 22° indicated that in amorphous particles, some small coherent regions were present. 

FTIR spectra showed that the surface properties of the nanoparticles differed considerably (Figure 2c). All tested nanoparticles exhibited a broad band ranging from 3200 to 3700 cm^−1^, which corresponded to the bridging H-bonded hydroxyls. This band was obviously overlapped with the sorbed water [19]. The peak centered at 3695 cm^−1^ belonged to free O–H vibrations [20], while the sharp peak at 3746 cm^−1^ represented “freely vibrating” surface hydroxyls in which each individual hydroxyl was sufficiently isolated to avoid interaction with the neighboring hydroxyls. For the coated and uncoated SiO_2_, the infrared group frequencies differed substantially. While SiO_2_–octyl was characterized by the stretching of alkyl –CH_3_ (2964 and 2857 cm^−1^) and –CH_2_ (2928 cm^−1^) bands, SiO_2_–methyl represented only –CH_3_ (2964 cm^−1^) stretching vibrations. In the case of SiO_2_, whose surface was not modified, these frequencies were not detected.

### 3.2. Evaluation of the Nanoparticles Cytotoxicity

The evaluation of the WST-1 (cell metabolism) and LDH (cell integrity) assays showed that the cytotoxicity of the nanoparticles differed substantially. While the ZnO and SiO_2_ nanoparticles exhibited statistically significant high-toxicity towards the A549 human lung cells, no statistically significant cytotoxic effect was observed for the coated SiO_2_ (–methyl, –octyl) and TiO_2_ nanoparticles, even up to their highest tested concentration of 250 μg L^−1^ (Figure 3). These results agreed favorably with the literature [21].

For the insoluble SiO_2_ nanoparticles, the LC_50_ was roughly 90 µg mL^−1^ for both cytotoxic assays (Table 3). Generally, this observation was in confirmation with previous studies showing SiO_2_ nanoparticles with induced cytotoxicity at concentrations of ˃25 µg mL^−1^ [22]. This could be explained by the formation of reactive oxygen species (ROS) inside the cells, due to Si–OH surface groups [8] or by membrane damage mediated by hydrogen bonding [23].

Alternatively, for the insoluble coated SiO_2_ (–octyl and –methyl) nanoparticles, no cytotoxicity towards the A549 cells was observed in both assays. This could be explained by the suppression of ROS formation, due to the presence of surface functionalities that inhibit the reactivity of the Si–OH groups. Other reasons include the surface modifications influencing nanoparticle-membrane interactions, intracellular trafficking, inter-particle interactions, or dynamic changes to the nanoparticle characteristics [24]. All of these factors can affect cell viability. 

Dose-dependent cytotoxicity was observed for the soluble ZnO nanoparticles, with LD_50_ being roughly 10 µg mL^−1^ for both assays. However, the cytotoxicity curve for the LDH assay was much less pronounced than the very sharp one for the WST-1 assay (Figure 3). This was probably due to the limited period for which the released LDH was present in the cell culture medium. After 24 h, the LDH enzyme might still have been inside the cells whose programmed cell death terminated with membrane disruption. Alternatively, the LDH enzyme might have been degraded within the 24 h, if cell death occurred shortly after exposure. No interference of the nanoparticles with the WST-1 and the LDH assay was observed. The discrepancy between the results obtained using the LDH and WST-1 assays shows the importance of employing cytotoxicity assays with different endpoints, to avoid underestimation of the results. Compared to the ZnO nanoparticles, the Zn^2+^ of ZnCl_2_ had a considerably higher cytotoxic effect, with LD_50_ being roughly 5 µg L^−1^. This was in agreement with the literature [25,26]. The lower cytotoxicity of the ZnO nanoparticles (8 µg L^−1^) could be explained by the gradual release of zinc ions from the internalized nanoparticles, as opposed to the complete dissolution of ZnCl_2_ in the cell culture. High concentrations of zinc ions can cause cell death through breakdown of the mitochondrial membrane potential [27].

For the insoluble TiO_2_ nanoparticles, no statistically significant cytotoxicity was observed. This observation was in agreement with published data. For instance, after 48 h exposure in dark, no cytotoxicity was observed towards A549 lung cells, up to a concentration of 400 µg mL^−1^ [14]. 

### 3.3. Nanoparticles Uptake by The Lung Cells

To improve the understanding of the cytotoxic test results, we employed transmission electron microscopy to investigate the localization of the nanoparticles within the lung cells. The TEM analysis of ultrathin sections of the cells revealed considerable differences in the localization of the internalized nanoparticles (Figure 4).

For the uncoated SiO_2_, while a majority of the nanoparticles was observed in the cell cytoplasm, only a small fraction of the nanoparticles was engulfed in the phagosomes. Along with intact phagosomes, partly disrupted ones were also observed, which presumably reflected nanoparticle release from phagosomes to cytoplasm. An increased production of autophagosomes engulfing the nanoparticle-contaminated cytoplasm and organelles was observed. This increase could be due to the need to degrade the internalized nanoparticles or the disrupted parts of the cells. Owing to the presence of the nanoparticles in the cytoplasm, the formation of reactive oxygen species could be hypothesized as a mechanism of their toxicity. 

SiO_2_-octyl and SiO_2_-methyl and TiO_2_ nanoparticles were dominantly internalized inside the intact phagosomes as aggregates of various sizes. Their intact membrane isolating the nanoparticles from the internal cellular milieu protected the cells from potential toxic effects. No morphological changes of the cell ultrastructure were observed, which was in agreement with our toxicological results. Very rarely, a small portion of the nanoparticles were observed in the cytoplasm, however, this could be an artifact of the sectioning procedure.

For ZnO, virtually all of the cells were damaged displaying the fragmented membranes and the nuclei. An increased number of autophagosomes and lipid droplets were found. As no nanoparticles were observed either inside or outside the lung cells, we hypothesized that their intracellular dissolution to Zn^2+^ had occurred. Velintine et al. (2017) observed the dissolution of ZnO nanoparticles, already 1 h after the cell exposure [28], which supported our hypothesis. Therefore, a cytotoxicity mechanism due to the Zn^2+^ dissolved in the cytoplasm is plausible. However, we cannot exclude the option that there was another mechanism that was caused by the nanoparticles themselves.

### 3.4. Improved Strength of Sandstone by Nanoparticle-Modified Consolidants

The drilling resistance forces of sandstone treated with various consolidants differed considerably. For instance, compared to the very low drilling resistance force for the reference sandstone (2 N), those of the consolidated sandstone samples were much higher (Figure 5, Table 4). For the commercial consolidant KSE OH, the increase in resistance was approximately three times higher (7 N). However, the highest drilling resistance forces were obtained for our novel nanoparticle-modified consolidants. The reason for the increase was most likely the improvement of the xerogel functional properties [2,29]. These include the suppression of the gel cracking and shrinking inside the stone and the increase of the xerogel hardness due to the embedding of nanoparticles. With regard to the effect of various nanoparticles on the consolidant performance, the highest resistance force was achieved for the TiO_2_ nanoparticles (26 N), while that of the ZnO nanoparticles was approximately 14 N. The resistance for both the coated particles and SiO_2_ was in between that of TiO_2_ and ZnO, ranging from ca. 17 to 22 N. Therefore, the SiO_2_ nanoparticles could be replaced with the coated ones, without any loss of mechanical performance. 

### 3.5. Study Significance and Limitations

The preliminary toxicological evaluations, using cytotoxicity as the basic toxicity endpoint, represented a time- and cost-effective approach in the design and development of novel materials and their applications. This kind of toxicological screening enables the selection and prioritization of materials for further development, therefore, minimizing the probability of investing additional resouces into the development of unsuitable materials with a high toxic potential and consequently a low practical applicability. In view of these considerations, the screening of in vitro toxicological assays, as presented in our study, plays an important role in nanotoxicology.

In vitro tests, however, do not take into account the complexity of multicellular organisms and processes occurring in specialized tissues and organs. Therefore, they might potentially lead to false-positive or false-negative conclusions. However, a thorough toxicological assessment using laboratory animals is extremely costly and time-consuming. Furthermore, due to the enormous variability of nanomaterials, it is not feasible to perform detailed toxicological studies for all types of nanomaterials that are potentially suitable for a particular application. In the event of the introduction of a selected nanomaterial-based consolidant in restoration practice, a more detailed toxicological assessment will be required.

It has been shown that there are significant differences in nanoparticle toxicity in different cell types. This indicates that the selection of the cell type might be an important factor affecting the results. However, no single cell line has, as yet, been recognized as a benchmark for the cytotoxicity evaluation of nanoparticles. The A549 cell line was selected for use in this study, being one of the most widely used airway cell models, due to its simple and cheap cultivation method and availability. These are important factors for screening toxicity testing. There are other potential cell lines, especially macrophages which are also suitable for cytotoxicity evaluation, as they represent a first-line biological barrier for nanomaterials in an organism and exhibit a high sensitivity. However, their sensitivity to endotoxin contamination, required for in vitro differentiation and lower homogeneity, might render them less suitable for screening testing, compared to A549 cells.

## 4. Conclusions

Using two independent WST-1 and LDH assays, we have shown that the toxicity of various nanoparticles differed considerably. While ZnO and SiO_2_ exhibited substantial cytotoxic effects in the test conditions, the TiO_2_ and both of the coated SiO_2_ nanoparticles did not. TEM analysis of the cellular nanoparticle uptake helped to interpret the toxicological results. Due to the encapsulation of the TiO_2_, SiO_2_-octyl, and SiO_2_-methyl nanoparticles in the intact phagosomes, their effects on cell viability were minimalized. However, the presence of the SiO_2_ nanoparticles in the cell cytoplasm caused their cytotoxicity. On the other hand, the cytotoxicity of the ZnO nanoparticles was most likely caused by their dissolution in the acidic conditions inside the phagosomes. Compared to sandstone and sandstone treated with a commercial consolidant, our novel nanoparticle-modified consolidants increased the stone strength by ten- and three-times, respectively. Our research showed that the often-used toxic nanoparticles (SiO_2_, ZnO) could be replaced by non-toxic equivalents (coated SiO_2_, TiO_2_), without any performance impairment. Thus, the combination of toxicological study and material research enabled the formulation of novel consolidants, which not only overcame the performance of more commonly used consolidants but are also safer for human health and the environment.

## Figures and Tables

**Figure 1 nanomaterials-09-00968-f001:**
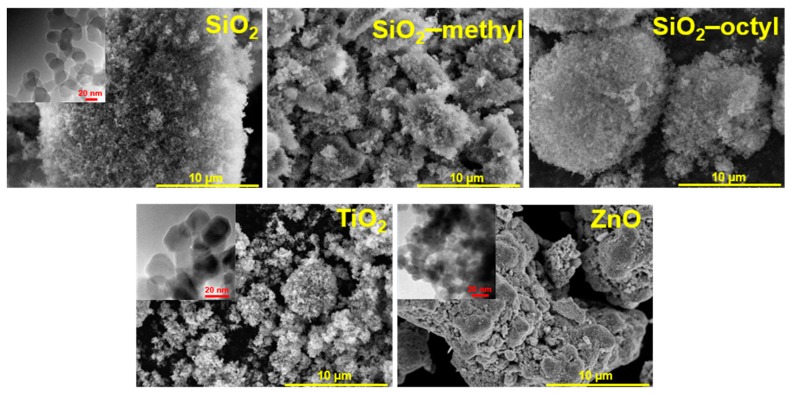
SEM images of the tested nanoparticles used for toxicological testing. The insets show details of the particles morphology determined by HRTEM.

**Figure 2 nanomaterials-09-00968-f002:**
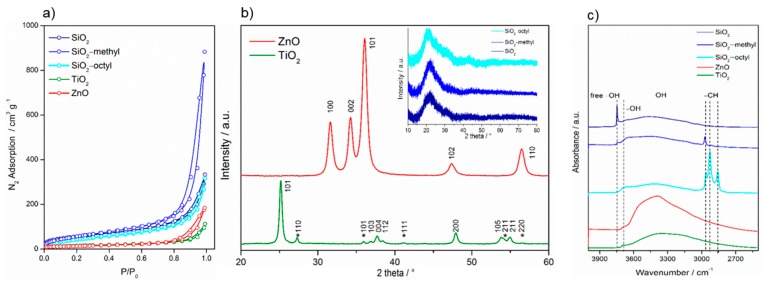
Nitrogen sorption isotherms of the nanoparticles determined at the boiling point of liquid nitrogen (**a**), XRD patterns of the nanoparticles (**b**), and FTIR spectra of the nanoparticles (**c**).

**Figure 3 nanomaterials-09-00968-f003:**
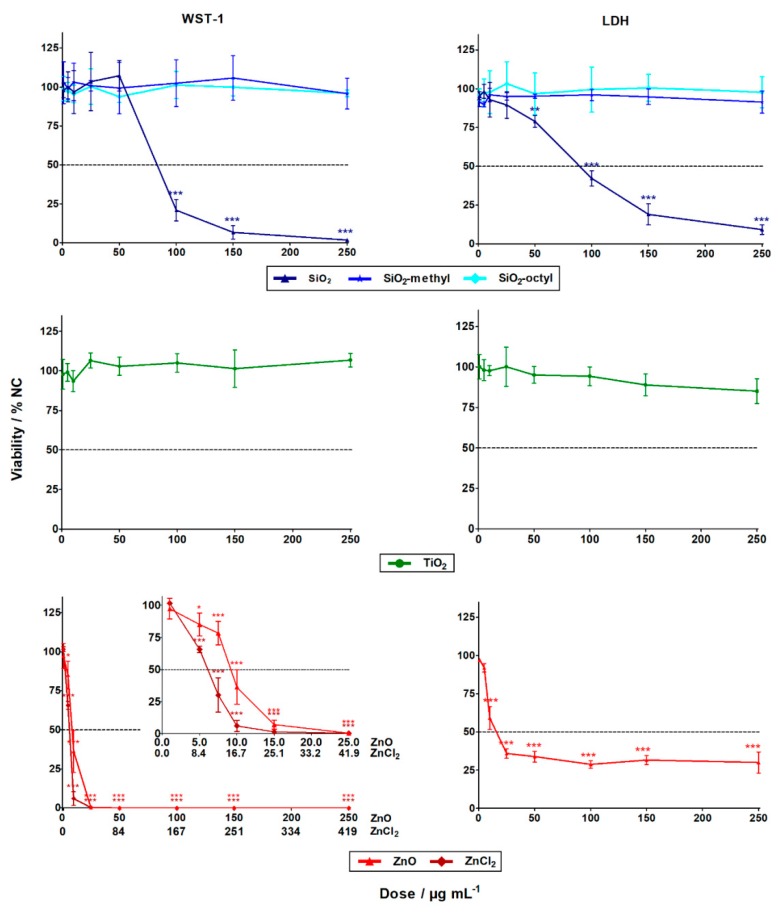
The cytotoxic effects of the nanoparticles and zinc salt (ZnCl_2_) in A549 cells after 24 h exposure determined by WST-1 (left column) and LDH (right column) assays. The data points designated with stars were statistically significant, *p* < 0.05 (*), *p* < 0.01 (**), and *p* < 0.001 (***) compared to the negative control cells (NC).

**Figure 4 nanomaterials-09-00968-f004:**
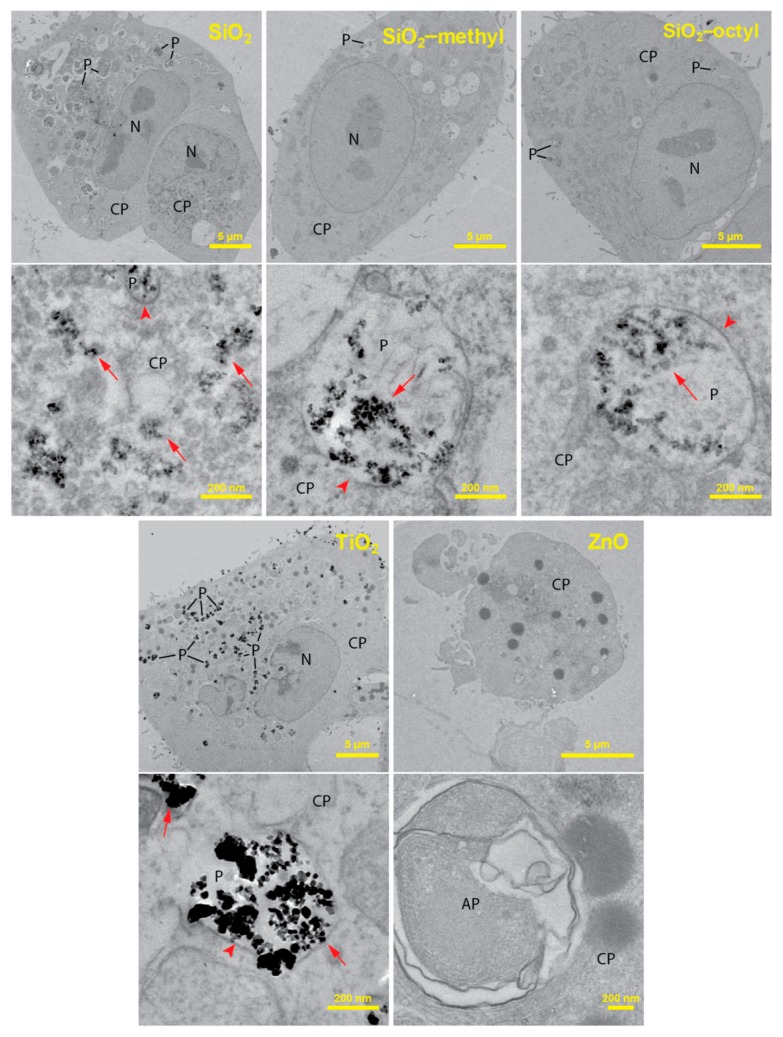
TEM cross-section images of the A549 cells after 24 h incubation with of TiO_2_, ZnO, SiO_2_, and modified SiO_2_ nanoparticles at a concentration of 10 μg mL^−1^. For each nanoparticle type in the pair of images, the upper represents the whole cell while the lower represents its high magnification detail depicting intracellular localization of the nanoparticles in the cytoplasm (CP) and the phagosomes (P). Nanoparticles were not observed in the autophagosomes (AP) and the cell nuclei (N). Arrows indicate nanoparticles; arrowheads show phagosomal membranes around nanoparticles.

**Figure 5 nanomaterials-09-00968-f005:**
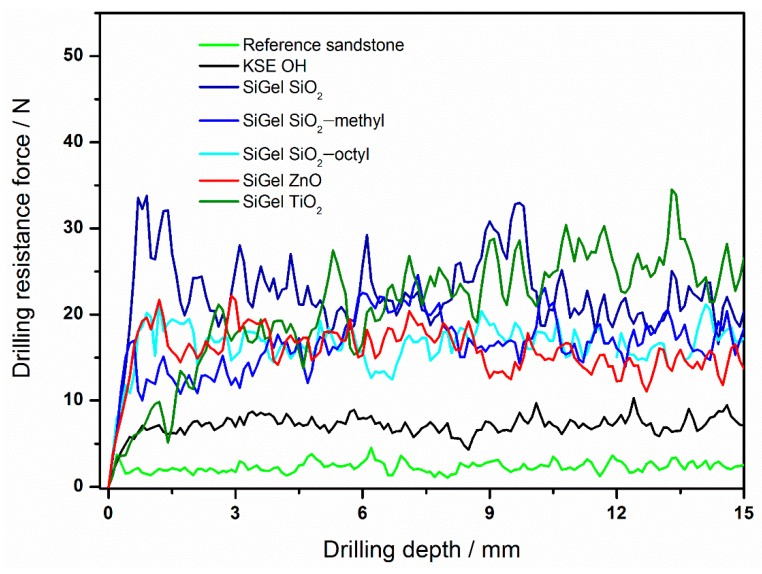
The drilling resistance profiles of the reference sandstone and sandstones treated with various consolidants.

**Table 1 nanomaterials-09-00968-t001:** An overview of the consolidants for the weathered sandstone.

Consolidant	Catalyst
KSE OH (commercial reference)	dibutyltin dilaurate
SiGel SiO_2_	n-octylamine
SiGel SiO_2_-methyl	n-octylamine
SiGel SiO_2_-octyl	n-octylamine
SiGel TiO_2_	n-octylamine
SiGel ZnO	n-octylamine + dibutyltin dilaurate

**Table 2 nanomaterials-09-00968-t002:** The properties of the nanoparticles used for cytotoxic testing.

Nanoparticles	Surface Modification	S_BET_ ^*^/m^2^ g^−1^	*C* ^*^	d ^*^/nm
SiO_2_	-	204	79	13
SiO_2_–methyl	–CH_3_	220	31	12
SiO_2_–octyl	–(CH_2_)_7_–CH_3_	164	27	16
TiO_2_	-	51	93	30
ZnO	-	46	139	23

^*^*S_BET_*—BET surface area; *C*—constant of the BET equation; *d*—particle size calculated from the S_BET_ provided the crystals were approximated by a sphere.

**Table 3 nanomaterials-09-00968-t003:** LC_50_ values obtained from both cytotoxic assays after 24 h exposure.

Sample	WST-1 Assay/µg mL^−1^	LDH Assay/µg mL^−1^
SiO_2_	89.4 ± 1.2	92.0 ± 12.4
SiO_2_–methyl	no toxic *	no toxic
SiO_2_–octyl	no toxic	not toxic
TiO_2_	no toxic	not toxic
ZnO	9.6 ± 0.2	9.5 ± 0.6
ZnO as Zn^2+^	7.8 ± 0.2	7.7 ± 0.5
ZnCl_2_	10.1 ± 0.6	NA
ZnCl_2_ as Zn^2+^	4.8 ± 0.3	NA

NA = not available. * Not toxic meant that the particles did not exhibit cytotoxic effects in the test conditions.

**Table 4 nanomaterials-09-00968-t004:** Drilling resistance force for sandstone consolidated with various consolidants. The average values in the table were determined within the drilling depth from four to fourteen mm.

Consolidant	Drilling resistance force/N
Reference sandstone	2.4 ± 0.7
KSE OH	7.4 ± 1.2
SiGel SiO_2_	22.3 ± 4.4
SiGel SiO_2_-methyl	16.9 ± 3.3
SiGel SiO_2_-octyl	17.3 ± 2.7
SiGel TiO_2_	25.9 ± 6.8
SiGel ZnO	14.4 ± 2.9

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
