# Peer review of "Toxicity of TiO2, ZnO, and SiO2 Nanoparticles in Human Lung Cells: Safe-by-Design Development of Construction Materials"

_nanomaterials, 2019, doi:10.3390/nano9070968_

Round 1

Reviewer 1 Report

In the study entitled "Toxicity of TiO2, ZnO and SiO2 nanoparticles in human lung cells: Safe-by-design development of construction materials” the authors compare toxicological effects of different nanoparticles (TiO2, ZnO, SiO2 and coated SiO2 nanoparticles).

Although the topic is interesting, in the latest years different papers present similar data about the same nanoparticles (except coated SiO2 nanoparticles). Moreover, the authors used only one cell line (A549 cells) to test the effects of the different nanomaterials and the cytotoxicity was evaluate only by WST-1 (metabolism) and LDH (cell integrity) assays. Therefore, the authors should improve the paper with other new experiments in order to differentiate their results from other data already published.

For example, the authors reported no cytotoxicity when A549 cells were treated with the insoluble SiO2 (-octyl and –methyl) nanoparticles and they explain this different behaviour by the suppression of ROS formation. It should be useful to demonstrate the reduced suppression of ROS by using a commercial kit (for example, using 2,7-dichlorodihydrofluorescein diacetate(H2-DCFDA)).

In addition, the authors should use other assays to evaluate the nanoparticles toxicity, such as TBARS (Thiobarbituric Acid Reactive Substances) production to evaluate lipid peroxidation.

Author Response

Comments and Suggestions for Authors

In the study entitled "Toxicity of TiO2, ZnO and SiO2 nanoparticles in human lung cells: Safe-by-design development of construction materials” the authors compare toxicological effects of different nanoparticles (TiO2, ZnO, SiO2 and coated SiO2 nanoparticles).

Comment: Although the topic is interesting, in the latest years different papers present similar data about the same nanoparticles (except coated SiO2 nanoparticles). Moreover, the authors used only one cell line (A549 cells) to test the effects of the different nanomaterials and the cytotoxicity was evaluate only by WST-1 (metabolism) and LDH (cell integrity) assays. Therefore, the authors should improve the paper with other new experiments in order to differentiate their results from other data already published.

Reply: The principal aim of our study was to perform the screening evaluation of the toxicity of nanoparticles with suitable technical properties. We believe that for this purpose using only one cell line is sufficient. We suppose that for nanoparticles negative within this first toxicity screening more detailed testing should follow, while nanoparticles positive at this stage should be eliminated from further testing. This approach will save efforts with testing many nanoparticles having similar functional properties but differing in their toxicity. Although it has been shown that there are significant differences in nanoparticle toxicity in different cell types, the trends in cytotoxicity tend to be the same across a wide range of cell types (i.e. the least or not cytotoxic nanoparticles exhibit low or no cytotoxicity in different cell types and highly cytotoxic nanoparticles have cytotoxic effects in different cell types;  see  e.g. Farcal et al. PloS one, 10(5), e0127174). The combination of evaluation of toxicity and functional properties is what differentiates this study from already published data.

Comment: For example, the authors reported no cytotoxicity when A549 cells were treated with the insoluble SiO2 (-octyl and –methyl) nanoparticles and they explain this different behaviour by the suppression of ROS formation. It should be useful to demonstrate the reduced suppression of ROS by using a commercial kit (for example, using 2,7-dichlorodihydrofluorescein diacetate(H2-DCFDA)).

Reply: We agree that information on ROS generation/suppression after nanoparticles treatment would be interesting. However, the aim of this study was to compare the cytotoxic potential of the selected nanoparticles not to determine mechanisms of the observed toxicity. Such mechanistic approach would be out of the scope of the study.

Comment: In addition, the authors should use other assays to evaluate the nanoparticles toxicity, such as TBARS (Thiobarbituric Acid Reactive Substances) production to evaluate lipid peroxidation.

Reply: This assay would complement the analysis of ROS generation discussed in the previous comment. Again, we agree with the reviewer that it would bring more details on the biological effects of the selected nanoparticles. However, the study design included the application of the basic tests of cytotoxicity followed by the development of the construction materials without going into the explanation of mechanisms of toxicity.

Reviewer 2 Report

There is no clear explanation as to why LDH is decreasing. It would be expected to increase in response to a toxin.

Author Response

Comments and Suggestions for Authors

Comment: There is no clear explanation as to why LDH is decreasing. It would be expected to increase in response to a toxin.

Reply: The concentration of LDH in the medium was actually increasing. However, the results are expressed as % viability so that a clear comparison with the results of the WST-1 assay could be provided - see the Method section line 140-141: The cell viability was calculated according to the following formula:

% viability = LDHlysates /(LDHlysates + LDHsupernatants) × 100

Reviewer 3 Report

The ms. by Remzova et al. evaluate the in vitro toxicity of 5 nanomaterials exploited to enhance the performance of construction materials. The results indicate that, while all the additives enhace the functional properties of the composite, their toxic effects in vitro widely vary, suggesting that additives that do not exhibit toxicity in vitro should be preferentially used in the productive process. I have several concerns:

a) From a biological point of view, the rank of toxicity described by the authors is expected from data present in literature (as recognized by the authors themselves). However, a preliminary assessment of in vitro toxicity would require the use of at least two distinct cell models, possibly of different lineage. It is true that A549 are a widely used airway cell model but also macrophage represent a first-line biological barrier for nanomaterials and their behavior should be assessed.

b) I do not think that the simple list of IC50 values at a single time point is enough. I would expect a graph showing the experimental data and a statistical assessment of the goodness of fit that yielded the parameters. Moreover, authors do no indicate how many doses they test and the single 24h experimental time may not be enough to exclude toxicity at later times.

c) The discussion of the difference between WST and LDH assays as far as ZnO is concerned (lines 258-260) is a bit strange since the curves are not actually shown. However, no mention is made in the methods of possible interference with LDH assay, while it is known that several ions can effectively inhibit the enzyme. Please, check if Zn inhibits LDH under the experimental conditions adopted.

d) More in general, authors should also discuss the possibility that in vitro tests do not reliably represent toxicity in vivo. This is a widely discussed issue in literature and, while their approach is of interest, they should not overlook the possibility that toxic effects detected in vitro are not confirmed in vivo or that (this would be worse...) negativity of a simple toxicity assay in vitro does not exclude toxic effects in vivo (for instance, genotoxicity or promotion of an inflammatory response).    

Author Response

Comments and Suggestions for Authors

The ms. by Remzova et al. evaluate the in vitro toxicity of 5 nanomaterials exploited to enhance the performance of construction materials. The results indicate that, while all the additives enhace the functional properties of the composite, their toxic effects in vitro widely vary, suggesting that additives that do not exhibit toxicity in vitro should be preferentially used in the productive process. I have several concerns:

Comment: a) From a biological point of view, the rank of toxicity described by the authors is expected from data present in literature (as recognized by the authors themselves). However, a preliminary assessment of in vitro toxicity would require the use of at least two distinct cell models, possibly of different lineage. It is true that A549 are a widely used airway cell model but also macrophage represent a first-line biological barrier for nanomaterials and their behavior should be assessed.

Reply: The principal aim of our study was to perform the screening evaluation of the toxicity of nanoparticles with suitable technical properties. We believe that for this purpose using only one cell line is sufficient. We suppose that for nanoparticles negative within this first toxicity screening more detailed testing should follow, while nanoparticles positive at this stage should be eliminated from further testing. This approach will save efforts with testing many nanoparticles having similar functional properties but differing in their toxicity. Although it has been shown that there are significant differences in nanoparticle toxicity in different cell types, the trends in cytotoxicity tend to be the same across a wide range of cell types (i.e. the least or not cytotoxic nanoparticles exhibit low or no cytotoxicity in different cell types and highly cytotoxic nanoparticles have cytotoxic effects in different cell types;  see  e.g. Farcal et al. PloS one, 10(5), e0127174).

Comment: b) I do not think that the simple list of IC50 values at a single time point is enough. I would expect a graph showing the experimental data and a statistical assessment of the goodness of fit that yielded the parameters. Moreover, authors do no indicate how many doses they test and the single 24h experimental time may not be enough to exclude toxicity at later times.

Reply: All the results are summarized in Figure 3, along with statistical significance and numbers of the tested doses. We added a list of the individual tested concentrations to the Methods section (line 106).

Comment: c) The discussion of the difference between WST and LDH assays as far as ZnO is concerned (lines 258-260) is a bit strange since the curves are not actually shown. However, no mention is made in the methods of possible interference with LDH assay, while it is known that several ions can effectively inhibit the enzyme. Please, check if Zn inhibits LDH under the experimental conditions adopted.

Reply: The discussion refers to Figure 3 in which a comparison between the WST-1 and LDH results is shown. The possible interference of the nanoparticles with the LDH assay was checked by the incubation of two highest nanoparticle concentrations (100 and 250 µg mL-1) with cell lysates (cells exposed to 1% Triton-X for 30 min) for 1 and 24 h before the LDH assay. No significant changes in the absorbance values (representing the LDH activity) were detected. Assessment of the nanoparticle interference with LDH assay was described in the Methods (141-144) and Results and Discussion (310-11) sections.

Comment: d) More in general, authors should also discuss the possibility that in vitro tests do not reliably represent toxicity in vivo. This is a widely discussed issue in literature and, while their approach is of interest, they should not overlook the possibility that toxic effects detected in vitro are not confirmed in vivo or that (this would be worse...) negativity of a simple toxicity assay in vitro does not exclude toxic effects in vivo (for instance, genotoxicity or promotion of an inflammatory response).   

Reply: The following paragraphs addressing the study’s significance and limitations were added in Results and Discussion (line 400-422):

The preliminary toxicological evaluations using cytotoxicity as the basic toxicological endpoint represent time- and cost-effective approach in the designing and development of novel materials and their applications. Such toxicological screening allows to select and prioritize materials for further development minimizing the probability of investing further resources into the development of materials with high toxic potential and consequently low practical applicability. For these reasons, screening in vitro toxicological assays, as presented in our study, play an important role in nanotoxicology.

In vitro tests, however, do not take into account the complexity of multicellular organisms and processes occurring in specialized tissues and organs. Thus, they may potentially lead to false positive or negative conclusions. On the other hand, thorough toxicological assessment using laboratory animals is extremely costly and time-consuming. Further, due to the enormous variability of nanomaterials, it is not feasible to perform detail toxicological studies for all types of nanomaterials that are potentially suitable for a particular application. In case of introduction of a selected nanomaterial-based consolidant in restoration practice, more detailed toxicological assessment will be required.

Round 2

Reviewer 1 Report

Even if different reviewers reported the same critical points, the authors tried to respond and this helped to improve the knowledge.

Author Response

The reviewer has not suggested any revision.

Reviewer 3 Report

 Original comment 1.

Authors state "We believe that for this purpose using only one cell line is sufficient" and cite the paper by Farcal et al. to support this conclusion. I disagree. Their conclusion would be justified if a single cell line were recognized as a benchmark for cytotoxicity, but this is not the case. If authors do not want to test the materials also on different cells, I think that, at least, they should recognize this limitation in the text.

Original comment 2.

OK

Original comment 3.

OK.

Original comment 4.

OK.

Please, check the text added since some sentences require revision. 

Author Response

 Original comment 1.

Authors state "We believe that for this purpose using only one cell line is sufficient" and cite the paper by Farcal et al. to support this conclusion. I disagree. Their conclusion would be justified if a single cell line were recognized as a benchmark for cytotoxicity, but this is not the case. If authors do not want to test the materials also on different cells, I think that, at least, they should recognize this limitation in the text.

Reply: The text was modified accordingly. See the last paragraph of the section 3.5.

Original comment 4.

OK.

Please, check the text added since some sentences require revision.

Reply: Text added was revised by a native speaker.